# A Novel *GUCA1A* Variant Associated with Cone Dystrophy Alters cGMP Signaling in Photoreceptors by Strongly Interacting with and Hyperactivating Retinal Guanylate Cyclase

**DOI:** 10.3390/ijms221910809

**Published:** 2021-10-06

**Authors:** Amedeo Biasi, Valerio Marino, Giuditta Dal Cortivo, Paolo Enrico Maltese, Antonio Mattia Modarelli, Matteo Bertelli, Leonardo Colombo, Daniele Dell’Orco

**Affiliations:** 1Department of Neurosciences, Biomedicine and Movement Sciences, Section of Biological Chemistry, University of Verona, 37134 Verona, Italy; amedeo.biasi@univr.it (A.B.); valerio.marino@univr.it (V.M.); giuditta.dalcortivo@univr.it (G.D.C.); 2MAGI’S Lab s.r.l., 38068 Rovereto, Italy; paolo.maltese@assomagi.org (P.E.M.); matteo.bertelli@assomagi.org (M.B.); 3Department of Ophthalmology, ASST Santi Paolo e Carlo Hospital, University of Milan, 20142 Milano, Italy; antoniomattia.modarelli@gmail.com; 4MAGI Euregio, 39100 Bolzano, Italy

**Keywords:** GUCA1A, phototransduction, cone dystrophy, guanylyl cyclase, photoreceptors, neuronal calcium sensor, retinal degeneration, calcium binding proteins

## Abstract

Guanylate cyclase-activating protein 1 (GCAP1), encoded by the *GUCA1A* gene, is a neuronal calcium sensor protein involved in shaping the photoresponse kinetics in cones and rods. GCAP1 accelerates or slows the cGMP synthesis operated by retinal guanylate cyclase (GC) based on the light-dependent levels of intracellular Ca^2+^, thereby ensuring a timely regulation of the phototransduction cascade. We found a novel variant of *GUCA1A* in a patient affected by autosomal dominant cone dystrophy (adCOD), leading to the Asn104His (N104H) amino acid substitution at the protein level. While biochemical analysis of the recombinant protein showed impaired Ca^2+^ sensitivity of the variant, structural properties investigated by circular dichroism and limited proteolysis excluded major structural rearrangements induced by the mutation. Analytical gel filtration profiles and dynamic light scattering were compatible with a dimeric protein both in the presence of Mg^2+^ alone and Mg^2+^ and Ca^2+^. Enzymatic assays showed that N104H-GCAP1 strongly interacts with the GC, with an affinity that doubles that of the WT. The doubled IC_50_ value of the novel variant (520 nM for N104H vs. 260 nM for the WT) is compatible with a constitutive activity of GC at physiological levels of Ca^2+^. The structural region at the interface with the GC may acquire enhanced flexibility under high Ca^2+^ conditions, as suggested by 2 μs molecular dynamics simulations. The altered interaction with GC would cause hyper-activity of the enzyme at both low and high Ca^2+^ levels, which would ultimately lead to toxic accumulation of cGMP and Ca^2+^ in the photoreceptor outer segment, thus triggering cell death.

## 1. Introduction

The phototransduction cascade in photoreceptors is the first signaling event initiating vision, permitting the conversion of the energy carried by light and absorbed by the opsins in the photoreceptor outer segment into a chemical signal, namely the transient drop in the release of glutamate from the synaptic term, which is sensed by downstream neurons [1]. The extraordinary capability of phototransduction to kinetically adapt to a broad variety of light conditions relies on the fine regulation of the cascade by two second messengers, namely Ca^2+^ and cyclic guanosine monophosphate (cGMP). Subtle alterations of their levels in the outer segment during the response to light triggers feedback mechanisms, which permit a timely shutoff of the cascade as well as adaption to specific light or dark conditions [1,2]. Absorption of light by (rhod)opsin activates phosphodiesterase 6, which catalyzes the hydrolysis of cGMP, thereby causing its dissociation from cyclic nucleotide-gated channels (CNG) and their closure. The light-independent extrusion of Ca^2+^ from the Na^+^/Ca^2+^, K^+^-exchanger further adds to the drop of Ca^2+^ concentration in the photoreceptor outer segments below 100 nM in the light [3] and contributes to the hyperpolarization of the cell membrane, which propagates to the photoreceptor synaptic terminal. 

Subtle changes in Ca^2+^ concentration are promptly detected by the neuronal calcium sensors Guanylate Cyclase-Activating Proteins (GCAPs). Two isoforms (GCAP1 and GCAP2) are expressed in human rods and cones, but only GCAP1 seems to be actively involved in the phototransduction cascade as a modulator of retinal guanylate cyclase (GC) activity, GCAP2 being probably involved in other biochemical processes [4,5]. In the dark, the Ca^2+^-loaded GCAP1 adopts a conformation that prevents the activation of GC, thereby inhibiting the synthesis of cGMP. The light-induced drop in Ca^2+^ concentration induces the replacement of Ca^2+^ ions for Mg^2+^ in the same binding sites in GCAP1 [6,7]. The conformation adopted by Mg^2+^-GCAP1 stimulates the accelerated synthesis of cGMP by GC, thus permitting rapid restoration of dark-adapted cell conditions by reopening of the CNG channels [8,9]. Although two isoforms of retinal GC have been found in photoreceptors, namely GC1 (or RetGC-1, GC-E) and GC2 (RetGC-2, GC-F), the latter produces less than 30% of cGMP in murine retina [10]; therefore, the major player in phototransduction is GC1—which we will refer to as GC throughout this paper.

The gene coding for GCAP1, named *GUCA1A*, has been associated with autosomal dominant cone dystrophies (adCOD) [11,12,13,14,15,16,17,18,19,20,21,22,23,24,25], a class of severe retinal degeneration diseases characterized by central vision loss, impaired color vision, and photophobia [26]. More than twenty point-mutations in *GUCA1A* have been found to be pathogenic and the highly heterogeneous phenotype seems to be strictly related to the specific amino acid substitution; indeed, different side chains at the same position can lead to dramatically different biochemical properties at the protein level [27]. 

In this work, we identify a novel variant of GCAP1 associated with adCOD, resulting in the substitution of Asn 104 with the positively charged residue His (p.N104H) within the high affinity binding site EF-hand 3 (EF3), a highly conserved region among vertebrates. The same position has been previously associated with adCOD in two independent studies, where a single amino acid replacement (p.N104K) [20] and a double amino acid substitution (p.N104K and p.G105R) have been detected in two different families [28]. Clinical data based on a long follow-up of 16 years for the oldest patient point to a relatively slow progression: the boundaries of the lesion remained confined within the macula, with no sign of rod system involvement. To unveil the perturbed mechanisms in the signaling cascade at a molecular level, we expressed N104H-GCAP1 in a heterologous system and characterized its structural and functional properties through a thorough integration of biochemical and biophysical studies with molecular dynamics simulations. We found that, despite the significant loss of Ca^2+^ sensitivity, the novel GCAP1 variant activates the target GC at higher levels as compared to the wild type (WT), both under conditions of low and high intracellular Ca^2+^. The interaction of the novel GCAP1 variant with GC is tighter than that of its WT counterpart and induces a constitutive activity of the cyclase at physiological levels of Ca^2+^. Structural alterations induced by the N104H mutation are minor at all levels of GCAP1 structural organization, but they are enough to alter the allosteric communication with the N-terminal lobe.

## 2. Results

### 2.1. Clinical Phenotype and Disease Progression

#### 2.1.1. Patient 1

The proband was an 18-year-old female. She came to our attention at the age of 12 because her father, here referred to as patient 2, was affected by COD. She only complained mild photophobia. At the first visit her visual acuity (VA) was 1.0 in both eyes. At the last follow-up, at the age of 18, VA was 0.9 in both eyes. Refractive error was −3.00 sph −1.75 cyl/95 in her right eye and −2.00 sph −1.50 cyl/60 in her left eye. Ophthalmoscopic examination did not show any sign of macular affection. Fundus autofluorescence (FAF) revealed a mild perifoveal hyper-autofluorescence in both eyes (Figure 1), while optical coherence tomography (OCT) imaging showed irregularities of hyper-reflective outer retinal bands, with the line corresponding to interdigitation zone not clearly detectable (Figure 2). Anatomically, the foveal lesion did not show any detectable sign of progression during the 7-year follow-up. 

Full-field ERG examination showed characteristically reduced cone single-flash responses and a normal implicit time in the flicker amplitudes. The patient’s rod function was normal: in *GUCA1A*-related adCOD generally rod system remains preserved, although in some cases progression may occur over time (Figure 3). 

#### 2.1.2. Patient 2

Patient 2, the proband’s father, was 46 years old at the time of the observation. He complained photophobia and a decrease in VA from the age of 15. At the age of 25, his VA was 0.2 in both eyes. At his last follow-up, at the age of 41, his VA was 0.05 in both eyes, with also a −0.75 sph refractive error in both eyes. The fundus examination showed normal optic disks and retinal vessels with signs of foveal atrophy. Fundus autofluorescence (FAF) revealed macular hypo-autofluorescence circumscribed by a hyper-autofluorescent ring in each eye (Figure 1), while optical coherence tomography (OCT) imaging showed a marked reduction in central macular thickness and an atrophy of outer retinal layers (Figure 2). Full-field ERG examination showed normal scotopic responses and reduced responses for the photopic component (data not shown), in line with the diagnosis of COD. In patient 2 the macular lesion enlarged over time: we can thus hypothesize that his age and the longer time passed between the two follow-ups might have played a crucial role in the development of the condition, as compared to the clinical stability of the proband.

#### 2.1.3. Patient 3

Patient 3 is the 10-year-old brother of the proband and son of patient 2. He did not report symptoms related to macular disease. At the time of his first visit in our department (at the age of 10), VA was 0.9 in both eyes. Refractive error was −0.50 sph −1.50 cyl/15 in his right eye and −1.50cyl/180 in his left eye. Ophthalmoscopic examination did not show any sign of macular affection. Both FAF and OCT features resembled the alterations observed in the proband, as shown in Appendix A.

### 2.2. Identification of a Novel Variant of GUCA1A in Heterozygosis

The NGS genetic testing resulted in a mean coverage of targeted bases of 170.4X, with 97.4% covered at least 25X. We identified the novel heterozygous variant c.310A > C, p.(Asn104His) in the exon 4 of *GUCA1A* gene (NM_000409.4). With the help of the online tool VarSome [29] (accessed on 18 August 2021) the variant was classified as likely pathogenic, in accordance with the American College of Medical Genetics and Genomics guidelines (ACMG) [30], by the following scores:PM1: Located in a mutational hot spot and/or critical and well-established functional domain (e.g., active site of an enzyme) without benign variation.PM2: Absent from controls (or at extremely low frequency if recessive) in the Exome Sequencing Project, 1000 Genomes Project, or Exome Aggregation Consortium.PM5: Novel missense change at an amino acid residue where a different missense change determined to be pathogenic has been seen before. Alternative variant chr6:42146128 C⇒A (Asn104Lys) is classified as Likely Pathogenic, one star, by ClinVar (and confirmed using ACMG).PP2: Missense variant in a gene that has a low rate of benign missense variation and in which missense variants are a common mechanism of disease.PP3: Multiple lines of computational evidence support a deleterious effect on the gene or gene product (conservation, evolutionary, splicing impact, etc.).

Unfortunately, the proband’s father and brother were not available for genetic testing. However, the phenotypes of all three subjects and their autosomal dominant inheritance pattern are in line with the association to the *GUCA1A* gene.

### 2.3. Retinal Guanylate Cyclase Is Hyperactivated by N104H-GCAP1

The functional effects of the identified amino acid substitution were probed at the protein level by the heterologous expression and purification of the N104H-GCAP1 variant and by testing its ability to activate and inhibit GC in reconstitution experiments. We initially probed the functionality of N104H-GCAP1 by monitoring the regulation of GC at: (i) high Ca^2+^ levels, corresponding to the levels in dark-adapted photoreceptors; (ii) very low Ca^2+^ levels, such as in light-activated photoreceptors. The enzymatic activity, that is the rate of cGMP synthesis, was then compared with that of WT-GCAP1. N104H-GCAP1 showed an increased ability to activate GC in both conditions, as shown by the significantly higher cGMP production at both high (*p* < 0.001) and low (*p* < 0.001) Ca^2+^ concentration (Figure 4A). Despite the hyperactivation of GC observed at both Ca^2+^ levels, the relative activation capability (X-fold = 3.9, Table 1) was approximately half of that of the WT (X-fold = 7.4), which is indicative of a compromised ability of the GC-GCAP complex to switch between activated and inhibited. We then measured the apparent affinity (EC_50_) of GC for N104H-GCAP1 by evaluating the enzymatic activity as a function of the concentration of GCAP1 (Figure 4B). The affinity of the novel variant for GC (EC_50_ = 1.6 μM) was double that of the WT (EC_50_ = 3.2 μM; Table 1). Finally, we measured the Ca^2+^-dependence of GC-activity (IC_50_) for N104H-GCAP1 (Figure 4C and Table 1). The variant showed a clear inability to fully inhibit the GC target over the physiological range of Ca^2+^ variation (gray-shaded area): indeed, the IC_50_ corresponding to the activation profile (0.52 μM, Table 1) was doubled as compared to the one observed for WT-GCAP1 (0.26 μM). The cooperativity of the regulation process was slightly reduced (h = 1.77 for N104H vs. h = 2.04 for WT-GCAP1, Table 1).

### 2.4. Ca^2+^-Affinity Is Slightly Reduced in N104H-GCAP1

The N104H amino acid substitution affects the highly conserved residue 5 of the Ca^2+^-binding loop in EF3 (Figure 5). The carbonyl group constitutes a Ca^2+^-coordinator in Asn104 (Figure 5B, left) and it is lost in the N104H variant, which introduces a positively charged imidazole (Figure 5B, right), expected to perturb the Ca^2+^ coordination.

To experimentally assess the effects of the amino acid substitution we used sodium dodecyl sulphate-polyacrylamide gel electrophoresis (SDS-PAGE) and monitored the differential electrophoretic migration of GCAP1 variants upon ion binding. This method allows the assessment of conformational changes and highlights alterations in Ca^2+^ affinity of neuronal calcium sensors (NCS) [32]. Indeed, in the absence of ions NCS migrate with a mobility expected from their theoretical molecular weight (MW); however, upon Ca^2+^-binding their electrophoretic mobility exhibits a shift toward a lower apparent MW, which has been found to be proportional to their affinity [33]. Figure 6A reports the assay for WT-GCAP1 and for N104H-GCAP1. In line with previous studies [5], the mobility of WT-GCAP1 shifted from ~23 kDa in the absence of Ca^2+^ to ~17 kDa in the presence of Ca^2+^, without substantial alterations observed in the Mg^2+^-bound form compared to the apo-protein. The N104H variant showed a less prominent shift upon Ca^2+^-binding, suggesting a reduced affinity for Ca^2+^.

Differences in Ca^2+^-affinity of the novel variant as compared to the WT were quantitatively evaluated by an assay based on the competition for Ca^2+^ of the chromophoric Ca^2+^-chelator 5,5′Br_2_-BAPTA [10,11,34], whose absorption decreases upon ion binding. The pattern of Ca^2+^-titration of N104H-GCAP1 (Figure 6B) is typical of a protein that competes with the chelator, but whose affinity for Ca^2+^ is significantly reduced as compared to that shown by the WT (see Refs. [21,33] for typical titrations of WT-GCAP1). The individual macroscopic binding constants were all significantly reduced for N104H-GCAP1 and, accordingly, the apparent affinity for Ca^2+^ was approximately 20-fold lower than that of the WT (Table 1). Such a low affinity (K_d_^app^ = 11.2 μM) would make the novel variant unable to correctly regulate the GC in the physiological Ca^2+^-range (200–600 nM).

### 2.5. Protein Secondary, Tertiary and Quaternary Structure Are Slightly Affected by the N104H Mutation

Conformational changes in NCS proteins in response to ion binding can be conveniently studied by circular dichroism (CD) spectroscopy, which provides information on protein tertiary structure in the near UV range (250–320 nm) as well as on the secondary structure, in the far UV (200–250 nm). Moreover, by heating the system under controlled conditions and following the ellipticity at a fixed wavelength, CD can be used to assess protein thermal stability.

The far UV CD spectrum of N104H-GCAP1 (Figure 7A) showed the typical features of an all α-helix protein with two minima at 222 and 208 nm. Only subtle differences in the spectral shape were detected when comparing the variant with the WT. The ratio between the minima at 222 and 208 nm (θ_222_/θ_208_) is a valuable descriptor of the spectral shape. While this ratio slightly increased in both variants when switching from the apo- to the Mg^2+^-bound form (Table 2), no change could be detected for the N104H variant upon addition of Ca^2+^ (θ_222_/θ_208_ = 0.92). Characteristic of the mutant was also the slightly lower value of relative variation in ellipticity at 222 nm upon ion binding (∆θ/θ), which was 5.5% as compared to the 7.7% of the WT. The analysis of near UV CD spectra (Figure 7B) showed virtually no difference between WT and N104H-GCAP1 in the apo or Mg^2+^-bound forms (see Ref. [13] for WT spectra). The only slight difference was a completely negative band in the Tyr-Trp region (275–310 nm), shown by the mutant upon addition of Ca^2+^, which is at odds with the WT spectrum displaying positive dichroism in the 274–290 nm range. Overall, results from CD spectroscopy suggest very minor structural rearrangements of GCAP1 following the N104H substitution in all cation-bound states.

To investigate whether the N104H mutation could alter the exposure of GCAP1 to proteases, both variants were digested using trypsin in the absence and in the presence of Mg^2+^ and Ca^2+^ (Figure 7C). The time dependence of the proteolytic digestion of WT-GCAP1 in the presence of EDTA (Appendix A), EGTA and Mg^2+^ (Appendix A), or Mg^2+^ and Ca^2+^ (Appendix A) highlighted a clear stabilizing effect of Ca^2+^. Indeed, 60 min after initiating the proteolytic digestion some traces of undigested protein were still visible. A comparison between WT and N104H-GCAP1 after 10 min of proteolysis showed essentially the same pattern in the apo-variants (Figure 7C), while WT-GCAP1 showed a slightly higher stability in the Mg^2+^-bound form compared to N104H, as shown by the higher intensity of undigested bands. A similar proteolytic pattern was observed in the presence of Ca^2+^ for the two variants, with proteolytic fragments of bigger MW compared to the Mg^2+^-bound case. 

We corroborated limited proteolysis by thermal stability studies by monitoring the CD signal at 222 nm, corresponding to the minimum in the spectrum displaying the largest variation upon ion addition. The analysis of thermal denaturation profiles is reported in Table 2. In its apo-form, N104H-GCAP1 was 6 °C less stable than WT (T_m_ = 48.1 °C vs. 54.1 °C, Table 2). In line with proteolysis experiments, Mg^2+^ stabilized the structure of the mutant (T_m_ = 53.9 °C) less than that of the WT (T_m_ = 58 °C). Ca^2+^-binding significantly stabilized both variants, and no clear folded-to-unfolded transition could be detected under the experimental conditions (Figure 7D). The percentage of unfolding was however higher for N104H-GCAP1 (38%) as compared to WT (30%); Table 2).

WT-GCAP1 forms functional dimers under physiological conditions [35,36]. We tested whether this was the case also for N104H-GCAP1 by running analytical gel filtration (Figure 8A) and DLS (Figure 8B) measurements. Analytical gel filtration showed that both in the presence of Mg^2+^ alone and in the co-presence of Mg^2+^ and Ca^2+^ the protein elutes as a dimer, as indicated by the very similar apparent MW (42.9 kDa and 41.7 kDa, respectively; Table 2). This is substantially in line with what has been previously observed for WT-GCAP1 in the same experimental conditions (Table 2) and confirms that the N104H substitution does not modify the oligomeric state of GCAP1. A similar hydrodynamic diameter was measured by DLS for both variants, although the Mg^2+^-bound form of N104H-GCAP1 showed a small (0.6 nm) but significant (*p* < 0.001) increase in hydrodynamic radius with respect to the WT. 

The time evolution of the mean count rate (MCR) provides useful information to monitor the stability and colloidal properties of the protein dispersion. We thus monitored the MCR profile over 22 h (Figure 8B) for N104H-GCAP1. Interestingly, both the Mg^2+^ and Ca^2+^-bound forms of the GCAP1 variant did not show any aggregation propensity, but rather displayed regular and wide MCR oscillations, with a period of approximately 15 h. The largest oscillations were observed in the presence of Ca^2+^, which also showed a higher average MCR (~250 kcps) as compared to the Mg^2+^-bound form (~100 kcps). These wide oscillations, not commonly observed for GCAP1 variants, did not imply a clear change in the oligomeric state of N104H-GCAP1, as proven by the plot of the intensity of the main light scattering peak over the 22 h time frame (Figure 8D), which shows high stability in both conditions. Interestingly, while the average hydrodynamic diameter was in line with that reported in Table 2 and substantially the same applied for the Mg^2+^- and Ca^2+^-bound forms, data were more scattered in the presence of Mg^2+^, despite the smaller oscillations observed in the relative MCR profile. 

### 2.6. Exhaustive Molecular Dynamics Simulations Show Altered Structural Flexibility for N104H-GCAP1 in Different GC1-Activating States 

CD spectroscopy did not show any major structural rearrangement of the COD-associated GCAP1 variant following the N104H substitution; however, both the altered sensitivity for Ca^2+^ and the dysregulation of GC activity induced by the mutant suggest that subtle alterations may occur at the atomic level. We therefore ran exhaustive (2 μs) comparative molecular dynamics (MD) simulations of WT and N104H-GCAP1 with two Mg^2+^ ions or three Ca^2+^ ions bound, corresponding to the GC-activating and GC-inhibiting state, respectively. In line with the spectroscopic data, MD simulations did not highlight major structural rearrangements for the variant in any tested state (Figure 9); however, the analysis of protein structural flexibility, as described by the Cα Root-Mean Square Fluctuation, (RMSF) highlighted an altered flexibility of the N104H-GCAP1 backbone as compared to the WT in both signaling states. In detail, significantly higher flexibility of the exiting helix of EF3 and of the unoccupied ion-binding loop of EF4 was observed for the mutant in its Mg^2+^-bound form; in addition, both Mg^2+^ ions bound to EF2 and EF3 showed higher RMSF as compared to the WT. This finding, together with the overall higher RMSF detected throughout the protein sequence (Appendix A) is in line with the lower thermal stability observed for the variant (Table 2 and Figure 7D) in the Mg^2+^-bound form, and it is indicative of an allosteric effect exerted by the variant. Interestingly, a significant alteration of backbone flexibility was observed also in the Ca^2+^-bound state (Figure 9). The C-terminal domain was essentially more rigid in N104H-GCAP1 as compared to the WT; in particular, the mutation stabilized the transient helix (residues 120–135) connecting EF3 and EF4. The only exception is represented by the higher flexibility (i.e., lower stability) of the Ca^2+^ ions bound to EF3 and especially to EF4, thus in the EF-hand adjacent to that where the amino acid substitution occurred. Very interestingly, the 2 μs simulations highlighted a major increase in flexibility of the N-terminal domain, as clearly shown by the higher RMSF of the entering helix of EF1 (Appendix A), where residues interacting with the GC are located [37,38,39]. This is a purely allosteric effect of the N104H mutation, affecting a distal site in the protein with crucial functional properties.

## 3. Discussion

A clear association between *GUCA1A* and adCOD was established over 20 years ago [15] and in the last years the number of point mutations found in the same gene has significantly raised [11,12,13,14,15,16,17,18,19,20,21,22,23,24,25]. Mutations are biochemically heterogeneous; therefore, a detailed molecular analysis is needed to infer general genotype–phenotype relations. We presented a complete clinical and biochemical characterization of the novel N104H-GCAP1 variant associated with adCOD in an Italian family with three affected members. Missense mutations in Asn104 associated with COD were found in two previous studies, which makes the genotype–phenotype comparison especially intriguing. The first mutation (N104K) was found in two members of the same family, for whom double-flash ERGs showed significantly delayed rod recovery from an intense flash, which were attributed to dominant-negative effects that slowed the stimulation of GC [20]. Similarly to our study, Jiang et al. [20] reported a case with a long follow-up (12 years vs. 16 years in our study). Both data showed that rod system remained fairly unaffected during the follow-up while the disease - as evaluated by means of FAF in our case and ffERG in their case—showed signs of progression at the levels of cones. 

Results from gel-shift experiments and limited proteolysis in N104K-GCAP1 [20] were very similar to those observed in this study for N104H, which might indicate similar biochemical features. However, despite the alike physicochemical characteristics of the substituted side chain (a lysine or a histidine, in both cases a positively charged residue at neutral pH), some peculiarities emerged. A first noticeable difference regards regulation of GC activation by the two GCAP1 variants. While WT-GCAP1 exhibited an IC_50_ value (~250 nM) compatible with the physiological Ca^2+^ range, the N104K substitution shifted the IC_50_ to a ~3-fold higher value, in contrast to the ~2-fold shift observed for N104H in this study. Most importantly, the apparent affinity of N104H for GC was double that of the WT-GCAP1 (Figure 4B and Table 1), at odds with N104K, which showed reduced capability to activate GC and thus required more GCAP1 to achieve a similar activation level [20]. It should be noted, however, that reconstitution experiments in Ref. [20] were performed with murine GC1 and human GCAP1, and that species-dependent biochemical characteristics may exist for the GCAP-GC signaling complex [4,5].

Very recently, some of us identified an isolated case of COD where the patient carried a double *GUCA1A* mutation affecting Asn104 and the adjacent Gly105 (N104K-G105R), thus introducing two positively charged side chains (lysine and arginine, respectively) [28]. The clinical phenotype was significantly different as compared to both N104K [20] and N104H-GCAP1, and quite unusual. Indeed, severe alterations of the ERG were observed under both scotopic and photopic conditions, with abnormally attenuated b-wave components and a negative pattern not observed in other COD patients [28]. At the protein level, Ca^2+^-sensitivity was severely reduced, and the variant constitutively activated both human GC1 and GC2, although the X-fold value was 80-fold lower compared to the WT for GC1, and 18-fold lower for GC2. This is a major difference with N104H-GCAP1, which showed a less perturbed (approximately halved) X-fold compared to WT (Table 1). We should point out, however, that for N104H-GCAP1-stimulated GC1 the absolute levels of cGMP at both high and low [Ca^2+^] were significantly higher than the corresponding values for the WT (Figure 4A), at odds with the N104K-G105R double mutant, which also induced constitutive activation of GC1, but lower cGMP synthesis at low Ca^2+^ [28]. The cyclase is therefore hyperactivated by N104H-GCAP1 under conditions that mimic both dark- and light-adapted photoreceptors. In this respect, the effects of the N104H-GCAP1 substitution on the photoreceptor physiology could be similar to those observed in other GCAP1 variants, and cell degeneration could be attributed to the dysregulation of the homeostasis of second messengers, which may accumulate in the photoreceptor outer segment due to the constitutive activation of GC [40], thus leading to toxic effects attributed to both Ca^2+^ and cGMP [41,42]. The peculiar ERG response from the patient with the double the N104K-G105R substitution suggests a perturbation of the transmission to downstream neurons and points to a perturbation of the GCAP1-GC1 macromolecular complex at the photoreceptor synaptic terminal [13]. The ERGs observed in this study for N104H-GCAP1 are instead in line with those observed in prior COD patients, and do not suggest significant alterations of the synaptic processes.

In conclusion, we found that point mutations in the same position (Asn104) of the *GUCA1A* gene not only lead to clinically different phenotypes, but also generate distinct molecular phenotypes despite the absence of major structural alterations observed in any case (this study and previous ones [20,28]). Mutation-specific sensitivity toward cations, subtle alterations of protein stability in the presence of Mg^2+^ and Ca^2+^, specific alteration of protein flexibility in distinct signaling states and the dependence of GCAP1 dimerization on the presence of specific cations [36] all point to a very complex molecular scenario, in which focusing on the effects of mutations on individual proteins might be of little use in advancing the molecular understanding of disease. Instead, unveiling the molecular details of the protein–protein and protein–ion interactions involved in the altered signaling cascade should be the final goal to achieve a molecular-level understanding of the extremely heterogenous retinal dystrophies, including COD, and would constitute a solid basis for designing effective therapeutic interventions.

## 4. Materials and Methods

### 4.1. Clinical and Ophthalmological Examinations

Patients were enrolled at the Retinal Dystrophy Unit of ASST Santi Paolo e Carlo Hospital, University of Milan (Italy). They periodically undergo detailed clinical examination, including best-corrected visual acuity (BCVA), slit-lamp examination, spectral-domain optical coherence tomography (SD-OCT), fundus autofluorescence (FAF), and dark- and light-adapted full field electroretinogram (ffERG).

### 4.2. Genetic Testing

The proband’s DNA was extracted from whole blood with a commercial kit (Blood DNA Kit E.Z.N.A.; Omega Bio-Tek Inc., Norcross, GA, USA) and analyzed by targeted Next-Generation Sequencing (NGS) on the Illumina MiSeq instrument, using the PE 2× 150 bp protocol (Illumina, San Diego, CA, USA). Raw sequencing data generated by the NGS platform were analyzed using an in-house pipeline, as described elsewhere [43]. The custom gene-targeted panel comprises of 140 genes associated with non-syndromic retinal dystrophies. The identified variants were subsequently evaluated in compliance with the ACMG standards and guidelines for the interpretation of sequence variants [30], with the help of the human genomic variant search engine VarSome (https://varsome.com, accessed on 18 August 2021) [29]. Genetic testing was performed as part of the diagnostic routine and the proband was invited to sign an informed consent form after pre-test genetic counseling.

### 4.3. Cloning, Protein Expression and Purification of N104H-GCAP1

The cDNA of wild-type human GCAP1-E6S (Uniprot entry: P43080) was cloned into a pET-11a vector using NdeI and NheI restriction sites (Genscript). The E6S mutation was inserted to obtain the consensus sequence for post-translational N-terminal myristoylation by *S. cerevisiae* N-Myristoyl transferase (yNMT) [44]. Sequence variant c.310A>C p.(Asn104His) was introduced by site-directed mutagenesis using the aforementioned pET-11a-GCAP1-E6S plasmid (Genscript) as template. Heterologous expression of GCAP1 variants was performed in BL21 *E. coli* cells that were previously co-transformed with pBB131-yNMT. The protein was extracted from the inclusion bodies after denaturation in 6M Guanidine-HCl and renatured by dialysis against 20 mM Tris-HCl pH 7.5, 150 mM NaCl, 7.2 mM β-mercaptoethanol buffer. The refolded protein was finally purified after size exclusion chromatography (SEC, HiPrep 26/60 Sephacryl S-200 HR, GE Healthcare), followed by anionic exchange chromatography (AEC, HiPrep Q HP 16/10, GE Healthcare) as previously described [13,44], except for using AEC buffers at pH 8. Protein concentration after purification was assessed by Bradford assay [45] using a GCAP1-specific reference curve based on amino acid hydrolysis assay (Alphalyze). Protein purity was evaluated on a 15% SDS PAGE gel. GCAP1 variants were exchanged against decalcified 50 mM NH_4_HCO_3_ buffer and lyophilized or against 20 mM Tris-HCl pH 7.5, 150 mM KCl, 1mM DTT buffer with three dialysis cycles (1 L each), and flash-frozen with liquid nitrogen. Samples were stored at −80 °C.

### 4.4. Guanylate Cyclase Enzymatic Activity Assays

To test whether the N104H substitution in GCAP1 affected the regulation of GC1 activity, specific enzymatic assays were conducted to monitor cGMP synthesis. HEK293 cells were used to stably express human recombinant ROS-GC1 (GC) as previously described [46]. GC assays were performed on isolated membranes obtained after cell lysis (10 mM HEPES pH 7.4, Protease Inhibitor Cocktail 1X, 1 mM DTT), 20 min incubation on ice, 20 min centrifugation at 18,000 × *g*, and resuspension of pelleted membranes in 50 mM HEPES pH 7.4, 50 mM KCl, 20 mM NaCl, 1 mM DTT buffer. Minimal and maximal GC activities were determined by incubating 5 µM of each GCAP1 variant with 2 mM K_2_H_2_EGTA (GC1-activating buffer) or K_2_CaEGTA (GC1-inhibiting buffer), leading, respectively, to <19 nM and ~30 µM free Ca^2+^ conditions. Enzymatic reactions were carried out in 30 mM MOPS/KOH pH 7.2, 60 mM KCl, 4 mM NaCl, 1 mM GTP, 3.5 mM MgCl_2_, 0.3 mM ATP, 0.16 mM Zaprinast buffer and blocked with the addition of 50 mM EDTA and boiling. Samples were then centrifuged for 20 min at ~18,000× *g* at 4 °C. The Ca^2+^ concentration at which GC activity is half-maximal (IC_50_) was determined by incubation of 5 µM of N104H-GCAP1 variant with different free [Ca^2+^] in the <19 nM–1 mM range. The GCAP1 concentration at which GC activation is half-maximal (EC_50_) was measured by incubation of increasing concentrations of N104H-GCAP1 (0–10 µM) in the presence of <19 nM free Ca^2+^. The cGMP synthesized during the enzymatic reactions was quantified by means of HPLC using a C18 reverse phase column (LiChrospher 100 RP-18, Merck). Data are reported as the mean ± standard deviation of at least three data sets. The statistical significance of the differences between the maximal and minimal GC activation by WT- and N104H-GCAP1 was assessed by two-tailed *t*-test (*p* = 0.001).

### 4.5. Gel mobility Shift Assay and Limited Proteolysis

SDS-PAGE under denaturing conditions was performed on a 15% acrylamide gel to investigate cation-induced electrophoretic mobility changes of GCAP1 variants. The experiment was carried out diluting proteins to a concentration of 30 µM in 20 mM Tris-HCl pH 7.5, 150 mM KCl, 1 mM DTT and by adding 2 mM EDTA, 1 mM EGTA + 1.1 mM Mg^2+^ or 1 mM Mg^2+^ and 1 mM Ca^2+^. After a 5-min incubation at 25 °C, samples were boiled and loaded onto the gel. Electrophoresis was run for 50 min at constant voltage (200 V) and protein bands were revealed by Coomassie blue staining. Limited proteolysis was performed on 20 µM WT and N104H-GCAP1 in the same conditions as for the mobility shift assay, with the addition of 0.3 µM trypsin (Sigma-Aldrich, St. Louis, MO, USA) to each reaction mix. To evaluate the optimal incubation time with trypsin (Appendix A), the reaction for WT-GCAP1 was stopped at different time steps, leading to the choice of a 10-min incubation for the comparison of the proteolytic pattern of WT and N104H-GCAP1 shown in Figure 6.

### 4.6. Ca^2+^-Binding Assays

The Ca^2+^-binding ability of N104H-GCAP1 was evaluated by a competition assay with the chromophoric chelator 5,5′Br_2_-BAPTA as previously described [21,33,34]. Lyophilized proteins were dissolved in a carefully decalcified buffer (20 mM Tris-HCl pH 7.5, 150 mM KCl, 1 mM Mg^2+^, 1 mM DTT; residual concentration of Ca^2+^ after decalcification was 0.15–0.7 µM) containing ~25 µM 5,5′Br_2_-BAPTA. The absorbance at 263 nm was recorded upon sequential additions of 3 µM Ca^2+^ to the solution at room temperature until a plateau was reached. Data were fitted to a three-sequential binding site model using CaLigator software [31] to estimate the individual macroscopic association constants (logK_i_) and apparent affinity constants (K_d_^app^ = 10^-(logK^ _1_^+ logK^ _2_
^+ logK^ _3_^)/3^) reported in Table 1, presented as average ± standard deviation of 5 technical replicates. Data shown in Figure 5 were normalized as follows to account for the total number of Ca^2+^ binding sites:(1)$$Normalized\;\left[Ca^{2+}\right]=\frac{\left[Ca^{2+}\right]}{\left[Q\right]+3\;\left[P\right]}$$
(2)$$Normalized\;y=\frac{A_{263}-A_{min}}{A_{max}-A_{min}}$$
where [*Q*] and [*P*] are the concentrations of 5,5′Br_2_-BAPTA and GCAP1 variants, respectively, measured at the end of each repetition by Bradford assay, *A*_263_ is the absorbance at 263 nm, *A_min_* and *A_max_* are the minimal and maximal absorbance values registered.

### 4.7. Circular Dichroism Spectroscopy and Thermal Denaturation Studies

The alterations in thermal stability, secondary and tertiary structure of N104H-GCAP1 upon ion binding were analyzed by means of Circular Dichroism (CD) spectroscopy on a Jasco J-710 spectropolarimeter supplied with a Peltier-type cell holder. Proteins were resuspended in 20 mM Tris-HCl pH 7.5, 150 mM KCl, 1 mM DTT and each recorded spectrum was the average of 5 accumulations. Far-UV CD spectra and thermal denaturation profiles were recorded in a 0.1-cm quartz cuvette with a protein concentration of 15 µM and 10 µM, respectively, in the presence of 300 µM EGTA, 300 µM EGTA + 1 mM Mg^2+^ or 1 mM Mg^2+^ + 300 µM free Ca^2+^. Near-UV CD spectra were recorded in a 1 cm quartz cuvette with a protein concentration of ~39 µM after serial additions of 500 µM EGTA, 1 mM Mg^2+^ and 500 µM free Ca^2+^. Thermal denaturation profiles were recorded by monitoring the ellipticity at 222 nm in a temperature window spanning from 20 °C to 96 °C (scan rate 90 °C/h). Denaturation data were fitted according to the following model, as in [21]:(3)$${𝛳}_{222}=\frac{\left(b_{n}+k_{n}T\right)+\left(b_{u}+k_{u}T\right)\;exp\left[-\Delta G_{nu}\left(T\right)\right]}{1+exp\;exp\left[\frac{-∆G_{nu}\left(T\right)}{RT}\right]\;}$$
where *n* and *u* are the native and unfolded states, *b* is the baseline value, *T* is the temperature, *k* is the slope of the plateaus and Δ*G_nu_* is the Gibbs free energy for folded-to-unfolded transition which can also be expressed in terms of change in enthalpy and heat capacity upon denaturation at constant pressure as follows (4)$$∆G_{nu}\left(T\right)=-\left(\Delta H\left(1-\frac{T}{T_{m}}\right)\right)+\Delta C_{p}\left(T-T_{m}-Tln\frac{T}{T_{m}}\right)$$

### 4.8. Analytical Gel Filtration

Analytical gel filtration was employed to analyze the apparent molecular weight and the oligomeric state of GCAP1 variants under Mg^2+^ and Ca^2+^-saturating conditions. Protein samples (20 µM) were loaded onto a Superose 12 10/300 column (GE Healthcare), previously equilibrated with 20 mM Tris-HCl pH 7.5, 150 mM KCl, 1 mM DTT + 500 µM EGTA and either 1 mM Mg^2+^ or + 1 mM Mg^2+^ and 1 mM Ca^2+^ at room temperature. Elution profiles were collected at 280 nm and the distribution coefficient (D_c_) was calculated as follows:(5)$$D_{c}=\frac{V_{e}-V_{v}}{V_{t}-V_{v}}$$
where *V_e_* is the elution volume, *V_t_* represents the total column volume (25 mL) and *V_v_* is the void volume (8 mL). Finally, the molecular weight was estimated from the calibration curve of log (MW) vs. *D_c_* as previously described [47].

### 4.9. Dynamic Light Scattering Analyses

To investigate the variations in hydrodynamic diameter, oligomeric state, and aggregation propensity of N104H-GCAP1 in different cation-loading states, the samples from aSEC were directly analyzed in a Zetasizer Nano-S (Malvern Instruments, Malvern, UK) at 25 °C using previously established settings [48]. The analysis of the hydrodynamic diameter and of the mean count rate, representing the time evolution of the colloidal properties of the suspension, was carried out for 22 h (~450 measurements, each averaging 13–15 repetitions). The hydrodynamic diameter reported in Table 2 is the mean of the first 30 measurements ± standard error of the mean (s.e.m.).

### 4.10. Protein Modeling and Molecular Dynamics Simulations

The homology model of Ca^2+^-loaded myristoylated human GCAP1 (UniProt entry: P43080) was obtained employing the “Advanced Homology Modeling” tool provided by the software Bioluminate (Maestro package v. 12.5.139, Schrödinger), using the Ca^2+^-loaded myristoylated GCAP1 from *G. Gallus* (PDB entry: 2R2I [49]) as a template. The N104H mutation was performed in silico in the obtained human homology model using the “Mutate Residue” tool by selecting the most probable rotamer. The activating form of human GCAP1 (Mg^2+^-bound) was obtained by deleting the Ca^2+^ ion bound to EF-4 and substituting the remaining Ca^2+^ ions in EF-2 and EF-3 with Mg^2+^ as performed in earlier work [50]. Molecular Dynamics (MD) simulations were run on the GROMACS 2020.3 package [51] using the all-atom CHARMM36m [52] force field, implemented with the parameters of the N-terminal myristoylated Gly (available upon request). Both Mg^2+^-bound and Ca^2+^-loaded GCAP1 variants underwent energy minimization and equilibration procedures as elsewhere described (2 ns in NVT ensemble with and without position restraints) [53] prior to the production phase, which consisted of two independent 1 µs replicas at constant pressure (1 atm) and temperature (310 K) for each state. Protein flexibility was assessed by monitoring the Root-Mean Square Fluctuation of Cα, that is the time-averaged Root-Mean Square Deviation as compared to the average structure of the concatenated 2 µs trajectories, following the analysis of consistency between the replicas [53] based on the root-mean square inner product (RMSIP, Appendix A) of the first 20 principal components (calculated on Cα) representing the largest collective motion of the protein.

## Figures and Tables

**Figure 1 ijms-22-10809-f001:**
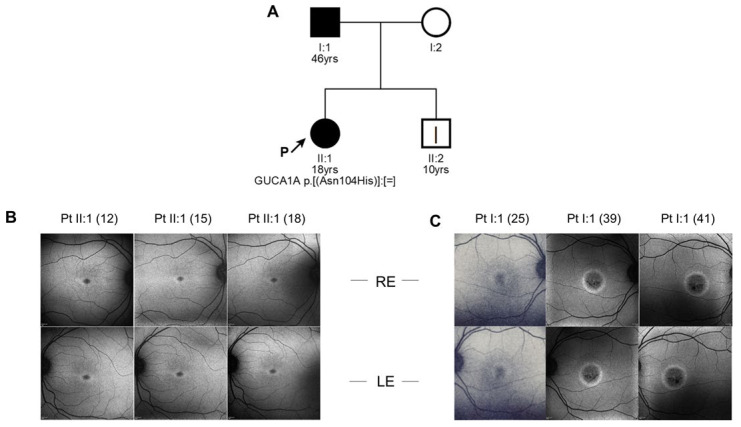
Clinical data and mutation inheritance. (**A**) Pedigree showing the matching genotype–phenotype segregation of the mutant allele of *GUCA1A*, harboring the c.310A > C mutation causing the amino acid substitution N104H. Legend: square, male subject; circle, female subject; black symbol, affected subject; white symbol, healthy subject; yrs, subject age at his/her last clinical evaluation; P, proband; square brackets ([;];[;]), maternal and paternal chromosome; =, no change; |, sub-clinic phenotype. Fundus images of (**B**) patient II:1 (age 12, 15, 18) and (**C**) patient I:1 (age 25, 39, 41)’s right (upper panels) and left eyes (lower panels).

**Figure 2 ijms-22-10809-f002:**
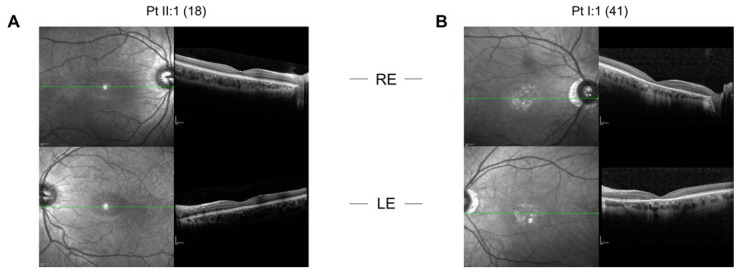
Spectral-domain optical coherence tomography (SD-OCT) scans of the right (upper panels) and left eyes (lower panels) of (**A**) the proband at the age of 18 and (**B**) of patient 2 I:1 at the age of 41. The green line indicates the interdigitation zone.

**Figure 3 ijms-22-10809-f003:**
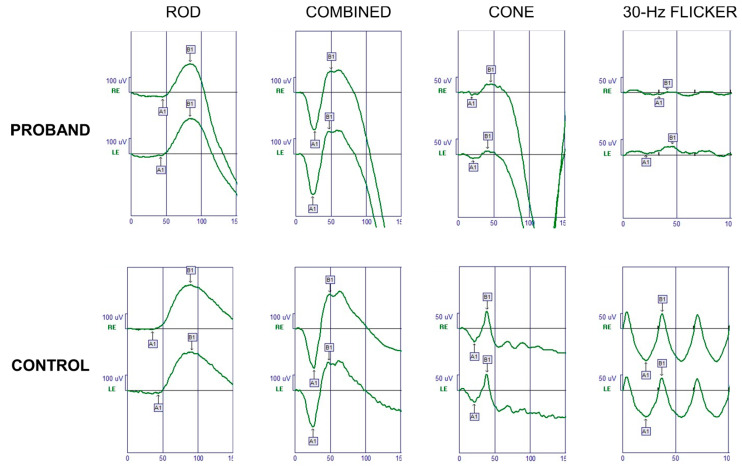
Full-field ERG of the proband’s right eye (RE) and left eye (LE). Normal responses are provided for comparison (Control). After a 30-min dark adaptation, dark-adapted responses (Rod and Combined) were within normal limits, while photopic components were reduced (Cone and 30-Hz flicker).

**Figure 4 ijms-22-10809-f004:**
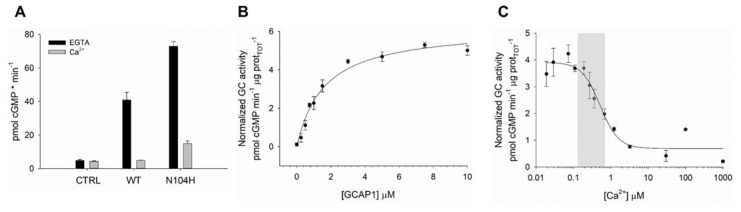
Ca^2+^-dependent Guanylate Cyclase (GC) regulation by GCAP1 variants. (**A**) Membranes containing GC were reconstituted with 5 µM WT or N104H-GCAP1 and <19 nM Ca^2+^ (black) or ~30 µM free Ca^2+^ (grey); control data were obtained using membranes without addition of GCAP1; reported data refer to average ± standard deviation of 6 technical replicates. (**B**) GC activity as a function of N104H-GCAP1 concentration (0 - 10 µM) in the presence of <19 nM Ca^2+^. (**C**) GC activity as a function of Ca^2+^ concentration (<19 nM - 1 mM) in the presence of 5 µM N104H-GCAP1. The physiological window of variation in Ca^2+^ concentration in photoreceptors is represented by the grey-shaded area. Measured enzymatic parameters are reported in Table 1.

**Figure 5 ijms-22-10809-f005:**
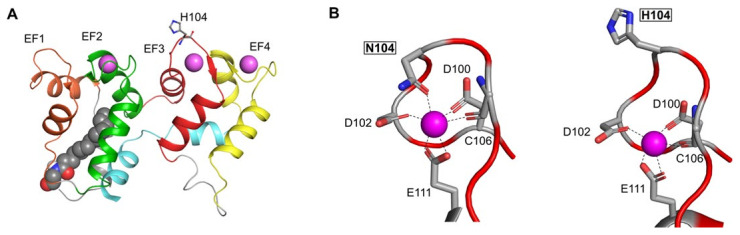
Structural model of N104H-GCAP1. (**A**) The 3D structural model of N104H-GCAP1 in the Ca^2+^-loaded form is depicted in the cartoon, with the N-terminal helix in grey, EF1 in orange, EF2 in green, EF3 in red, EF4 in yellow, the C-terminal helix in cyan. Ca^2+^ ions and the myristoyl group are represented as pink and grey spheres, respectively, and mutated residue H104 is represented as sticks and colored by element. (**B**) Ca^2+^ ion coordination is shown by key residues of EF3 in WT-GCAP1 (left) and N104H-GCAP1 (right). Structures depicted represent the last frames extracted from the first replica trajectories of both variants.

**Figure 6 ijms-22-10809-f006:**
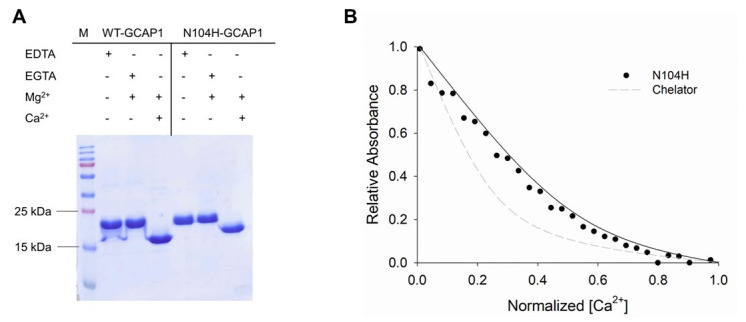
Ca^2+^-binding assays for N104H-GCAP1. (**A**) The 15% SDS-PAGE gel of 30 µM of WT and N104H-GCAP1 in the presence of 2 mM EDTA, 1 mM EGTA + 1.1 mM Mg^2+^, 1 mM Mg^2+^ and 1 mM Ca^2+^. (**B**) Example of a Ca^2+^-titration curve for N104H-GCAP1. The normalized absorption of 5,5′Br_2_-BAPTA in competition with N104H-GCAP1 upon Ca^2+^-titration in the presence of 1 mM Mg^2+^ is shown as black circles, together with data fitting to a three-Ca^2+^-binding site model using CaLigator [31] (black line), and the theoretical curve of the chelator in the absence of competition (grey dashed line).

**Figure 7 ijms-22-10809-f007:**
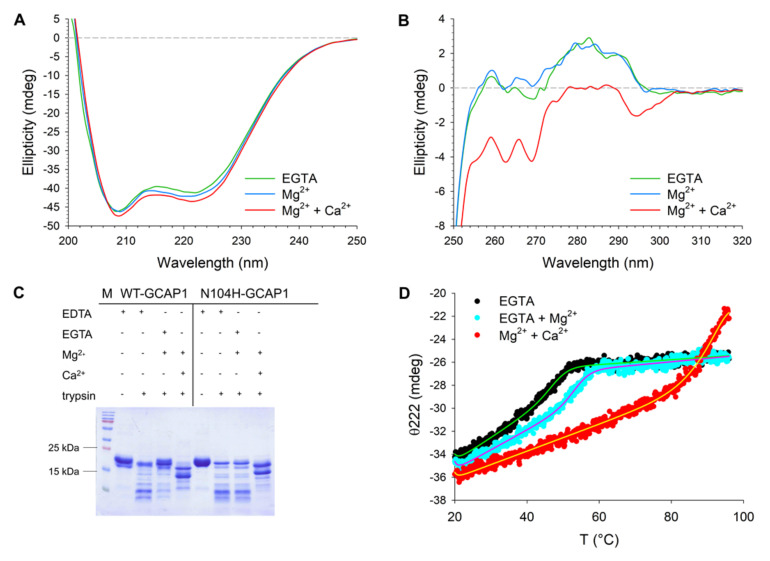
Structural and stability changes occurring in N104H-GCAP1 upon ion binding. (**A**) Far-UV CD spectra of 15 µM N104H-GCAP1 in the presence of 300 µM EGTA (green) and after serial additions of 1 mM Mg^2+^ (blue) and 300 µM free Ca^2+^ (red). (**B**) Near-UV spectra of ~39 µM N104H-GCAP1 in the presence of 500 µM EGTA (green) and after serial additions of 1 mM Mg^2+^ (blue) and 500 µM free Ca^2+^ (red). (**C**) Limited proteolysis of 20 µM WT and N104H-GCAP1 after 10 min incubation with 0.3 µM trypsin in the presence of 2 mM EDTA, 1 mM EGTA + 1.1 mM Mg^2+^ or 1 mM Mg^2+^ and 1 mM Ca^2+^. Variants in the presence of 2 mM EDTA and in the absence of trypsin represent the reference MW of the undigested protein. (**D**) Thermal denaturation profiles of 10 µM N104H-GCAP1 in the presence of 300 µM EGTA (black), 300 µM EGTA + 1 mM Mg^2+^ (blue) or 1 mM Mg^2+^ + 300 µM Ca^2+^ (red). CD spectroscopy measurements were carried out in 20 mM Tris-HCl pH 7.5, 150 mM KCl, 1 mM DTT buffer. Thermal denaturation profiles were collected by monitoring the ellipticity at 222 nm in a temperature range spanning from 20 °C to 96 °C and were fitted to a function accounting for thermodynamic contributions (see Methods section).

**Figure 8 ijms-22-10809-f008:**
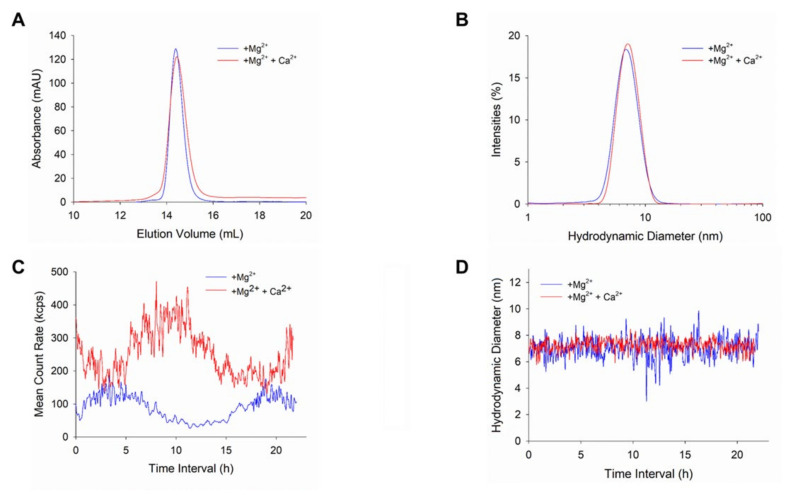
Analysis of N104H-GCAP1 quaternary structure. (**A**) Analytical gel filtration profile of ~45 µM N104H-GCAP1. Chromatographic runs were carried out in either 500 µM EGTA + 1 mM Mg^2+^ (blue) or 1 mM Mg^2+^ + 1 mM Ca^2+^ (red). (**B**) Hydrodynamic diameter of ~45 µM N104H-GCAP1 monitored by DLS in the presence of 500 µM EGTA + 1 mM Mg^2+^ (blue) or 1 mM Mg^2+^ + 1 mM Ca^2+^ (red). Solid lines represent the mean curve of 30 measurements. Time evolution over 22 h of the mean count rate (**C**) and (**D**) peak 1 mean intensity of ~45 µM N104H-GCAP1 in the presence of 500 µM EGTA + 1 mM Mg^2+^ (blue) or 1 mM Mg^2+^ + 1 mM Ca^2+^ (red).

**Figure 9 ijms-22-10809-f009:**
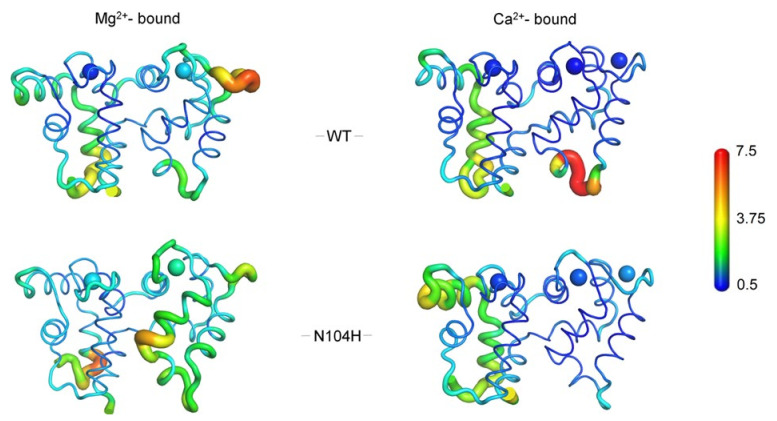
Cα Root-Mean Square Ffluctuation (RMSF) projected on the 3D structure of WT (upper panels) and N104H (lower panels) GCAP1 in their GC-activating Mg^2+^-bound (left panels) and inhibiting Ca^2+^-loaded (right panels) forms. Protein structure is displayed as tube cartoons, with radius proportional to the RMSF; Mg^2+^ and Ca^2+^ are depicted as spheres. Structures and ions are colored in a rainbow scheme representing RMSF values ranging from 0.5 to 7.5 Å (see Appendix A for Cα-RMSF profiles).

**Table 1 ijms-22-10809-t001:** Enzymatic regulation and Ca^2+^-affinity of GCAP1 variants. ^a^ [Ca^2+^] at which GCs activity is half-maximal; ^b^ Hill coefficient; ^c^ [N104H] at which GCs activity is half-maximal; ^d^ calculated as (GC_max_-GC_min_)/GC_min_, where GC_max_ and GC_min_ represents the maximal and minimal cGMP production; ^e^ decimal logarithm of the macroscopic Ca^2+^-binding constants after data fitting to a three independent binding sites model obtained with CaLigator [31]; ^f^ apparent affinity values calculated after averaging logK_i_; ^g^ WT data are taken from Ref. [27].

Variant	IC_50_ ^a^ (µM)	h ^b^	EC_50_ ^c^ (µM)	X-fold ^d^	logK_1_ ^e^	logK_2_ ^e^	logK_3_ ^e^	K_d_^app f^ (µM)
WT ^g^	0.26 ± 0.01	2.05 ± 0.21	3.2 ± 0.3	7.4	7.07 ± 0.13	5.55 ± 0.19	-	0.49
N104H	0.52 ± 0.1	1.77 ± 0.59	1.6 ± 0.2	3.9	5.92 ± 0.09	4.7 ± 0.48	4.23 ± 0.34	11.2

**Table 2 ijms-22-10809-t002:** Structural and stability descriptors extrapolated from CD spectroscopy, hydrodynamic diameter estimation by DLS, and apparent MW monitored by analytical SEC. ^a^ calculated as (ϴ_222_^ion^–ϴ_222_^EGTA^)/ϴ_222_^EGTA^; ^b^ melting temperature estimated by fitting ellipticity at 222 nm, as described in the Methods; ^c^ calculated as (ϴ_222_^96^
^°C^–ϴ_222_^20^
^°C^)/ϴ_222_^20^
^°C^; ^d^ average hydrodynamic diameter ± s.e.m.; ^e^ number of measurements used for hydrodynamic diameter calculations; ^f^ apparent MW estimated by analytical gel filtration; ^g^ data were taken from Ref. [27]; ^h^ data were taken from Ref. [13].

Variant	Condition	θ_222_/θ_208_	Δθ/θ ^a^ (%)	T_m_ ^b^ (°C)	Unfolding ^c^ (%)	d ^d^ (nm) [n] ^e^	MW ^f^ (kDa)
WT	EGTA	0.90 ^g^	-	54.1 ^h^	24.6 ^h^	-	-
	Mg^2+^	0.91 ^g^	2.8 ^g^	58 ^h^	30.8 ^h^	6.35 ± 0.07 ^g^ [27]	45.9 ^g^
	Ca^2+^	0.95 ^g^	7.7 ^g^	>96 ^h^	30.4 ^h^	6.85 ± 0.17 ^g^ [20]	47.8 ^g^
N104H	EGTA	0.89	-	48.1	24.9	-	-
	Mg^2+^	0.92	1.97	53.9	26.1	6.98 ± 0.1 [30]	42.9
	Ca^2+^	0.92	5.41	>96	38.0	6.97 ± 0.09 [30]	41.7

## Data Availability

The data reported in this work are available upon request from the corresponding authors and are not available to the public because of their size.

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
