# Peer review of "A Novel GUCA1A Variant Associated with Cone Dystrophy Alters cGMP Signaling in Photoreceptors by Strongly Interacting with and Hyperactivating Retinal Guanylate Cyclase"

_ijms, 2021, doi:10.3390/ijms221910809_

Round 1

Reviewer 1 Report

The authors conducted rigorous scientific experiments, evaluating multiple possibilities for interpreting the investigated phenomena. As a suggestion, the authors could increase the value of their work and potential impact by presenting concrete ways of protection, such as wearing specific glasses, to respond to the nature of macular degeneration specific to this pathology. With such an approach, the article would be easily interpreted by a wider range of researchers. 

Author Response

The authors conducted rigorous scientific experiments, evaluating multiple possibilities for interpreting the investigated phenomena. As a suggestion, the authors could increase the value of their work and potential impact by presenting concrete ways of protection, such as wearing specific glasses, to respond to the nature of macular degeneration specific to this pathology. With such an approach, the article would be easily interpreted by a wider range of researchers. 

We are glad that this Reviewer found merit in our work, and we thank the Rveiewer for the suggestion. We know from our previous published work* that blue-violet light-filtering artificial lenses may improve both visual acuity and contrast sensitivity by cutting a preponderant part of blue-violet glaring radiations, thus providing more favorable conditions for efficient visual signal coding from the residual photoreceptors; however, in this specific pathology there is no evidence that filtering lenses can help prevent disease progression. We therefore prefer not to mention protective devices in this specific case.

* L. Colombo, E. Melardi, P. Ferri et al. “Visual function improvement using photocromic and selective blue-violet light filtering spectacle lenses in patients affected by retinal diseases” – BMC Ophthalmogy - 2017 Aug 22;17(1):149

Reviewer 2 Report

Authors performed an extensive study on the adCOD disease-mutation association. I have some minor remarks listed below.

“Unfortunately, the proband’s father and brother were not available for genetic testing. However, the phenotypes of all three subjects and their autosomal dominant inheritance pattern..” - Authors describe here only three related patients. Although the disease was observed in all of them only in one of them the mutation was confirmed. Apart from it, it is too small dataset to drive conclusions based on it. The association mutation-disease should be also confirmed at least in several, non-related cases. Authors should also discuss and exclude other possible issues, e.g, gene regulatory networks or other inherited diseases.

Authors refer their results for the mutation-disease association to other data described in ref. 20. Yet, the actual genetic data there is again only two families’ members.

“Figure 1. Clinical and genetic data.” – but it actually shows only clinical data and comparison of the phenotype of only two related patients developing the same disease. It should be renamed to, e.g., ‘clinical data and inheriting of the mutation’.

A sentence from Discussion: “A clear association between GUCA1A and adCOD has been established over 20 years ago [15]” should be rather moved to Introduction.

There is a lot of similar expressions between the previous work of Authors (ref. 28) and the current one, e.g., in abstracts. Authors should refer to their previous work in the current one at the very beginning of the manuscript.

Regarding Fig. S3 – could Authors explain the most striking difference between mutant and WT in the region 120-140 (Ca2+-loaded) and 140-160 (Mg2+-bound) that is not close in the protein sequence to the described mutation?

“which introduces a positively charged imidazole (Figure 5C)” – where is Figure 5C?

Please, use rather a three-letters code of amino acids instead of “asparagine 104”.

Author Response

Authors performed an extensive study on the adCOD disease-mutation association. I have some minor remarks listed below.

We are glad that this Reviewer found merit in our work. We have addressed all the points raised as follows.

“Unfortunately, the proband’s father and brother were not available for genetic testing. However, the phenotypes of all three subjects and their autosomal dominant inheritance pattern..” - Authors describe here only three related patients. Although the disease was observed in all of them only in one of them the mutation was confirmed. Apart from it, it is too small dataset to drive conclusions based on it. The association mutation-disease should be also confirmed at least in several, non-related cases. Authors should also discuss and exclude other possible issues, e.g, gene regulatory networks or other inherited diseases.

We agree with this Reviewer that the association between mutation and disease cannot be directly assessed in the absence of segregation studies, which could not be performed in the absence of consent for genetic analyses of the other patients.  Nonetheless, several lines of experimental evidence from different groups (see references 11-28) verified the connection between mutations in GUCA1A gene and inherited retinal dystrophies, essentially cone and cone-rod dystrophies. In addition, two independent studies (see references 20 and 28) reported genetic evidence of the association of the same genic locus with a different substitution (N104K) in patients affected by cone dystrophy. Finally, the genetic analysis of the proband was performed by sequencing 140 genes associated with non-syndromic retinal dystrophies, which did not highlight other relevant variants.  We can thus rule out the involvement of other known genes in this case and consider these lines of evidence  sufficient to clearly establish a genotype-phenotype relationship in this study.

Authors refer their results for the mutation-disease association to other data described in ref. 20. Yet, the actual genetic data there is again only two families’ members.

The Reviewer is right. We point to the same arguments explained above for the mutation-disease association in this case. We would like to point out the difficulty in performing robust segregation studies in the absence of patient’s consent to sequencing.

“Figure 1. Clinical and genetic data.” – but it actually shows only clinical data and comparison of the phenotype of only two related patients developing the same disease. It should be renamed to, e.g., ‘clinical data and inheriting of the mutation’.

We thank the Reviewer for the suggestion and rephrased Figure 1 caption as follows:

Figure 1. Clinical data and mutation inheritance”

A sentence from Discussion: “A clear association between GUCA1A and adCOD has been established over 20 years ago [15]” should be rather moved to Introduction.

We believe that this sentence is functional to introduce the discussion of the results, therefore we prefer to keep it where it is. The same concept was more extensively addressed in the introduction (see paragraph “The gene coding for GCAP1, … , different biochemical properties at the protein level [27].”)

There is a lot of similar expressions between the previous work of Authors (ref. 28) and the current one, e.g., in abstracts.

We are not sure about what the Reviewer is referring to with “There is a lot of similar expressions between the previous work of Authors”. If our interpretation is correct, the similar expressions present in this work and in ref 28 are unavoidably similar, since the mutations described in the two manuscripts affect the same gene involved in the same physiological process. Furthermore, the investigation was carried out taking advantage of the same techniques with a similar molecular phenotype. Results are, however, significantly different.

Authors should refer to their previous work in the current one at the very beginning of the manuscript.

We are not sure about what the Reviewer is referring to, the previous work on the double mutation N104K-G105R is referred in the last paragraph of the introduction.

Regarding Fig. S3 – could Authors explain the most striking difference between mutant and WT in the region 120-140 (Ca2+-loaded) and 140-160 (Mg2+-bound) that is not close in the protein sequence to the described mutation?

The Reviewer is raising an interesting point. We clarified such differences by expanding the respective results section as follows:

“In detail, significantly higher flexibility of the exiting helix of EF3 and of the unoccupied ion-binding loop of EF4 was observed for the mutant in its Mg2+-bound form; in addition, both Mg2+ ions bound to EF2 and EF3 showed higher RMSF as compared to the WT. This finding, together with the generally higher RMSF detected throughout the protein sequence (Figure S3) is in line with the lower thermal stability observed for the variant (Table 2 and Figure 7D) in the Mg2+-bound form, and it is indicative of an allosteric effect exerted by the variant. Interestingly, a significant alteration of backbone flexibility was observed also in the Ca2+-bound state (Figure 9). The C-terminal domain was essentially more rigid in N104H-GCAP1 as compared to the WT, in particular the mutation stabilized the transient helix (residues 120-135) connecting EF3 and EF4. The only exception is represented by except for the higher flexibility (i.e. lower stability) of the Ca2+ ions bound to EF3 and especially to EF4, thus in the EF-hand adjacent to that where the amino acid substitution occurred.

“which introduces a positively charged imidazole (Figure 5C)” – where is Figure 5C?

We are sorry for the mistake and we corrected the text as follows:

“The N104H amino acid substitution affects the highly conserved residue 5 of the Ca2+-binding loop in EF3 (Figure 5). The carbonyl group constitutes a Ca2+-coordinator in Asn104 (Figure 5B, left) and it is lost in the N104H variant, which introduces a positively charged imidazole (Figure 5B, right), expected to perturb the Ca2+ coordination.”

Please, use rather a three-letters code of amino acids instead of “asparagine 104”.

We corrected the text as suggested.